# Novel Treatment Strategy Using Second-Generation Androgen Receptor Inhibitors for Non-Metastatic Castration-Resistant Prostate Cancer

**DOI:** 10.3390/biomedicines9060661

**Published:** 2021-06-09

**Authors:** Doo Yong Chung, Jee Soo Ha, Kang Su Cho

**Affiliations:** 1Department of Urology, Inha University School of Medicine, Incheon 22212, Korea; dychung@inha.ac.kr; 2Department of Urology, Prostate Cancer Center, Gangnam Severance Hospital, Yonsei University College of Medicine, Seoul 06273, Korea; engzsu@yuhs.ac

**Keywords:** second-generation androgen receptor inhibitors, non-metastatic castration-resistant prostate cancer, apalutamide, enzalutamide, darolutamide

## Abstract

Non-metastatic castration-resistant prostate cancer (nmCRPC) is defined by a progressively rising prostate-specific antigen level, despite a castrate level of testosterone, in the absence of obvious radiologic evidence of metastatic disease on conventional imaging modalities. As a significant proportion of patients with nmCRPC develop metastatic diseases, the therapeutic goals of physicians for these patients are to delay metastasis development, preserve quality of life, and increase overall survival (OS). Since 2018, the treatment of nmCRPC has changed dramatically with the introduction of second-generation androgen receptor inhibitors, such as enzalutamide (ENZA), apalutamide (APA), and darolutamide (DARO). These drugs demonstrated substantial improvements in metastasis-free survival (MFS) and OS in phase III randomized clinical trials. In addition, these drugs have an excellent safety profile, preserve quality of life, and can delay disease-related symptoms. A recently published indirect meta-analysis reported that APA and ENZA showed better findings in MFS and that DARO had relatively fewer adverse effects. However, in the absence of a direct comparison, careful interpretation is required. Thus, APA, ENZA, and DARO should be considered the new standard drugs for treating nmCRPC.

## 1. Introduction

Androgen deprivation therapy (ADT) [1], encompassing surgical and chemical castration [2], has shown significant clinical benefits in the management of advanced and metastatic prostate cancer [3]. Most prostate cancers initially respond to ADT but eventually develop resistance, transforming from castration-sensitive to castration-resistant prostate cancer (CRPC) [4]. Non-metastatic castration-resistant prostate cancer (nmCRPC) is a unique disease condition regarded as a “bridge” to metastatic CRPC (mCRPC). Continuous ADT is advised for nmCRPC until it progresses to metastatic disease [5]. Imaging studies have revealed the incidence of metastasis despite continuous ADT in patients with a median metastasis-free survival (MFS) of approximately 15 months [6,7]. First-generation androgen receptor inhibitors (ARi), such as bicalutamide or flutamide, and antiandrogen withdrawal syndrome therapy are conventionally used to treat nmCRPC but without any significant survival benefit [8,9,10,11]. In recent years, the development of and conduction of clinical trials for novel second-generation ARi have changed the treatment landscape for nmCRPC. Second-generation ARi bind to androgen receptors with a 7- to 10-fold greater affinity than do first-generation ARi [12,13,14]. In addition, the formers bind directly to the ligand-binding domain of the androgen receptor, inhibit its nuclear translocation, inhibit DNA binding, and impede androgen receptor-mediated transcription [15,16]. Between 2018 and 2019, the Food and Drug Administration (FDA) approved three second-generation ARi for nmCRPC [17]. These drugs are orally administered and their characteristics are summarized in Table 1. They raised new hopes of prolonging overall survival (OS) of patients with nmCRPC and changed the treatment paradigm for nmCRPC. In this review, we discuss and compare three second-generation ARi that have undergone large-scale phase 3 randomized clinical trials (RCTs).

## 2. Definition of nmCRPC

To assist in clinical decision making, six index patients were described in the American Urology Association guideline to represent the most common clinical scenarios; the disease condition of the first index patient was defined as asymptomatic, nmCRPC [25]. The National Comprehensive Cancer Network guidelines define nmCRPC as CRPC lacking evidence of metastases on conventional imaging [26]. More specifically, the Prostate Cancer Clinical Trials Working Group 3 defined nmCRPC as a progressively rising prostate-specific antigen (PSA) level, namely a 25% PSA increase from nadir (starting PSA ≥ 1.0 ng/mL), with a minimum increase of 2 ng/mL despite a castrate level of testosterone (<50 ng/dL) in the absence of obvious radiologic evidence of metastatic disease on conventional imaging modalities [27].

A substantial percentage of patients with nmCRPC develop metastatic lesions. Smith et al. reported that 46% of men with nmCRPC developed metastasis within 2 years [5,28]. Recently, Moreira et al. also reported that among nmCRPC patients, nearly 60% developed metastatic disease during the first 5 years, with most of the cases of metastasis occurring within the first 3 years [29]. In nmCRPC, baseline PSA level, PSA velocity, and PSA doubling time (PSADT) have been associated with time to bone metastases, MFS, and OS [30,31,32]. Therefore, the therapeutic goals of physicians in these patients are to delay metastasis development, preserve quality of life (QOL), and increase OS.

## 3. Phase III Clinical Trials Assessing the Effect of Second-Generation ARi for nmCRPC

### 3.1. SPARTAN (Apalutamide)

On 14 February 2018, the FDA approved apalutamide (APA; Erleada™, janssen Pharmaceuticals, Inc., Beerse, Belgium) for patients with nmCRPC. SPARTAN (NCT01946204) was a phase III, randomized, double-blind, placebo-controlled, multicenter study designed to evaluate the efficacy and safety of APA (240 mg per day) compared to those of placebo in 1207 patients (806 in the APA group and 401 in the placebo group) with high-risk nmCRPC (patients whose PSADT was ≤10 months) [7,33]. This study allowed the enrolment of patients with asymptomatic pelvic lymph node enlargement. Continuous ADT with a GnRH analog or surgical castration was required among the sample. The primary endpoint was MFS, which was defined as the time from randomization to the first detection of distant metastasis on imaging (as assessed by a blinded independent central review) or death from any cause, whichever occurred first.

The planned primary analysis was performed after 378 events (distant metastasis or death) had occurred [7]. Median MFS was 40.5 months in the APA group and 16.2 months in the placebo group (hazard ratio (HR) for metastasis or death, 0.28; 95% confidence interval (CI), 0.23–0.35; *p* < 0.001). The APA group showed improvements over the placebo group in all secondary endpoints, including time to metastasis (HR, 0.27; 95% CI, 0.22–0.34; *p* < 0.001) and progression-free survival (HR, 0.29; 95% CI, 0.24–0.36; *p* < 0.001). Time to symptomatic progression was significantly longer with APA than with placebo (HR, 0.45; 95% CI, 0.32–0.63; *p* < 0.001).

There was no statistically significant difference in the analysis of OS in the initial report (HR, 0.70; 95% CI, 0.47 to 1.04; *p* < 0.07) [7]. Thereafter, the final analysis for OS was reported when the median follow-up was 52 months [33]. By the time of this analysis, 428 deaths had occurred. After the primary endpoint analysis was met, the study was blinded. There were 76 (19%) patients in the crossover group who received APA after placebo. The median treatment duration was 32.9 months in the APA group, 11.5 months in the placebo group, and 26.1 months with APA in the crossover group. Subsequent life-prolonging therapy was received by 371 (46%) patients in the APA group and 338 (84%) patients in the placebo group, including crossover group patients. The median OS was 73.9 months in the APA group and 59.9 months in the placebo group (HR, 0.78; 95% CI, 0.64–0.96; *p* = 0.016). In addition, APA was associated with a significant prolongation of time to first cytotoxic chemotherapy (median not reached in either group; HR, 0.63; 95% CI, 0.49–0.81; *p* = 0.0002).

According to the final analysis, discontinuation rate due to progressive disease was 43% in the APA group and 74% in the placebo group. This was the most common cause of drug discontinuation. The rate of adverse events leading to discontinuation of the trial regimen was 15% (120/803) in the APA group and 7.3% (29/398) in the placebo group. Adverse effects (AEs) of grade 3 or 4 occurred in 56% (449/803) and 36% (145/398) of the patients in the APA and placebo groups, respectively. The AEs that occurred in more than 15% of patients in the APA group (vs. in placebo group) were as follows: fatigue (33% vs. 21%), hypertension (HTN) (28% vs. 21%), diarrhea (23% vs. 15%), fall (22% vs. 9.5%), nausea (20% vs. 16%), arthralgia (20% vs. 8.3%), weight loss (20% vs. 6.5%), back pain (18% vs. 15%), and hot flushes (15% vs. 8.5%).

### 3.2. PROSPER (Enzalutamide)

On 13 July 2018, the FDA approved enzalutamide (ENZA; Xtandi, Astellas Pharma US, Inc., Tokyo, Japan) for patients with nmCRPC. PROSPER (NCT02003924) was a phase III, randomized, double-blind, placebo-controlled (in a 2:1 ratio), multicenter study designed to evaluate the efficacy and safety of ENZA (160 mg per day) compared to those of placebo in 1401 patients with high-risk nmCRPC (patients whose PSADT was ≤10 months) [6,34]. This study included 933 patients in the ENZA group and 468 patients in the placebo group. Patients with suspected brain metastases, active leptomeningeal disease, a history of seizures, or a condition that may confer a predisposition to seizures were excluded.

The primary endpoint was MFS, defined as either the time from randomization to radiographic progression, as determined by a central reviewer at any time, or as the time to death from any cause during the period from randomization to 112 days after the discontinuation of the trial regimen without evidence of radiographic progression, whichever occurred first. In the primary analysis, a total of 219/933 patients (23%) in the ENZA group had metastasis or died as compared with 228/468 (49%) patients in the placebo group. The median MFS was 36.6 months in the ENZA group versus 14.7 months in the placebo group. A statistically significant and clinically meaningful improvement in MFS (HR, 0.29; 95% CI, 0.24–0.35; *p* < 0.001) was seen in the ENZA group. PSA progression-free survival was 37.2 months in the ENZA group and 3.9 months in the placebo group (HR, 0.07; 95% CI, 0.05–0.08; *p* < 0.001). The time to the first use of a subsequent antineoplastic therapy was longer with ENZA treatment than with placebo (39.6 months vs. 17.7 months; HR, 0.21; 95% CI, 0.17–0.26; *p* < 0.001). Similar to the SPARTAN study, the complete analysis of the OS was not released in the initial report. In subsequent reports [34], ENZA-treated patients showed a significant 27% decreased risk of death than placebo recipients did. The median follow-up period was 48 months, as of October 15, 2019. There were 466 deaths, of which 288/930 (30.9%) occurred in the ENZA group and 178/465 (38.0%) in the placebo group. A total of 87 patients received ENZA in the crossover group. ENZA significantly prolonged OS compared to placebo (HR, 0.73; 95% CI 0.61–0.89; *p* = 0.001). The median OS was 67.0 months (95% confidence interval (CI), 64.0 not reached (NR)) in the ENZA group and 56.3 months (95% CI, 54.4–63.0) in the placebo group. Subsequent antineoplastic therapies were initiated after treatment discontinuation in 310 (33%) patients in the ENZA group and 303 (65%) patients in the placebo group. Median duration of treatment was 33.9 months vs. 14.2 months with ENZA vs. placebo, respectively.

Based on the final analysis, grade ≥ 3 AEs were reported by 446/930 (48%) patients in the ENZA group vs. 126/465 (27%) patients in the placebo group. The rate of AEs leading to discontinuation of the trial regimen was 158/930 (17%) in the ENZA group and 41/465 (9%) in the placebo group. The AEs that occurred in more than 15% of patients in the ENZA group (vs. placebo group) were as follows: fatigue (46% vs. 22%); musculoskeletal events, including back pain, arthralgia, myalgia, musculoskeletal pain, pain in the extremities, musculoskeletal stiffness, muscular weakness, and muscle spasms (34% vs. 23%); hypertension (HTN) (18% vs. 6%); fall (18% vs. 5%); and fracture (18% vs. 6%).

### 3.3. ARAMIS (Darolutamide)

On 30 July 2019, the FDA approved darolutamide (DARO; Nubeqa, Bayer HealthCare Pharmaceuticals, Inc., Whippany, Hanover, NJ, USA) for nmCRPC treatment [35,36]. ARAMIS (NCT02200614) is the largest phase III randomized, double-blind, placebo-controlled trial conducted for nmCRPC to date. It included patients with a PSADT of ≤10 months and pelvic lymph nodes <2 cm below the aortic bifurcation. In this study, patients with previous seizures or conditions predisposing to seizures were permitted in both groups. This RCT assigned 1509 men, at a 2:1 ratio, to receive DARO (600 mg twice daily) (955 patients) or placebo (554 patients) while continuing to receive ADT. The final analysis was scheduled to be published after approximately 240 deaths occurred. However, the primary endpoint, MFS, was better in the DARO group, and the trial was changed to an unblinded study after only 136 deaths. OS and all other secondary endpoints were evaluated thereafter. Good OS was seen in the DARO group.

The median follow-up time was 29.0 months. At the time of data unblinding, all 170 patients who were still receiving the placebo started receiving DARO. One hundred and thirty-three patients who had discontinued placebo before data unblinding received at least one other life-prolonging therapy.

The MFS results from the ARAMIS trial are presented. In total, 221/955 patients (23%) in the DARO group developed metastasis or died, as compared with 216/554 (39%) patients in the placebo group. Participants treated with DARO achieved a median MFS of 40.4 months (95% CI, 34.3 NR), more than double the 18.4 months (95% CI, 15.5–22.3) achieved by those given placebo. This was a statistically strong significant finding (HR, 0.41; 95% CI, 0.34–0.5; *p* < 0.001). The OS rate at 3 years was 83% (95% CI, 80–86) in the DARO group and 77% (95% CI, 72–81) in the placebo group. The risk of death was significantly lower in the DARO group than in the placebo group (HR, 0.69; 95% CI, 0.53–0.88; *p* = 0.003). DARO was also associated with a significant benefit with respect to all other secondary endpoints, including the time to pain progression (HR, 0.65; 95% CI, 0.53–0.79; *p* < 0.001) that was extended by 14.9 months in the DARO group compared to that in the placebo group. The time to the first symptomatic skeletal event (HR, 0.48; 95% CI, 0.29–0.82; *p* < 0.001) and the time to the first use of cytotoxic chemotherapy were reported (HR, 0.58; 95% CI, 0.44–0.76; *p* < 0.001).

Grade 3–4 AEs were reported by 10.9% (104/954) of the patients in the DARO group vs. 6.5% (36/554) of the patients in the placebo group. No AEs were seen in more than 15% of the DARO group participants. The most common AE was fatigue (13% vs. 8%). Therefore, the trial reported that the incidence of AE after treatment initiation was similar between the two groups. In addition, the discontinuation rates were comparable between the placebo (8.9%) and DARO (8.7%) groups.

We summarized and presented the data in three studies in Table 2 and Table 3.

## 4. Comparison of Efficacy and Safety of Second-Generation ARi

Presently, the scope of FDA approval for the three drugs differs. APA is approved for nmCRPC and metastatic castration-sensitive prostate cancer (mCSPC) [7,33,37,38]. ENZA is an androgen receptor antagonist approved for all three indications: nmCRPC, mCSPC, and mCRPC [6,34,39,40]. Finally, DARO is an androgen receptor antagonist approved only for nmCRPC [36].

These ARi have been proven effective in phase III RCTs in patients with nmCRPC. Many similarities exist among the SPARTAN, PROSPER, and ARAMIS trials. All participants had high-risk nmCRPC, defined by a baseline PSA level of 2 ng/mL and a PSADT ≤10 months, and the primary endpoint in each trial was MFS as assessed by computed tomography and a bone scan of the pelvis, chest, and abdomen every 16 weeks. The three RCTs targeted high-risk patients with nmCRPC. Nodal disease was present in patients in all three trials. Although both the ARAMIS and SPARTAN trials permitted the enrolment of patients with malignant nodes <2 cm in diameter located below the aortic bifurcation, only the SPARTAN trial set a threshold of node size <1.5 cm. All patients underwent ADT throughout the intervention phase.

There was one difference between the participants in the three trials; patients with a history of seizures were excluded from the PROSPER and SPARTAN trials. This is because administration of ENZA or APA is associated with an increased risk of seizures [41,42,43] due to penetration of the compound through the blood–brain barrier (BBB) and subsequent inhibition of γ-aminobutyric acid receptors [44]. In contrast, DARO has limited penetration through the BBB and, thus, a limited effect on mental status. This has been demonstrated in preclinical trials of DARO [45,46]. A study showed that the penetration rate of ENZA and APA to the BBB was more than 10 times that of DARO [47]. Therefore, DARO has been reported to be safe in patients with a history of seizures [14], and, thus, the ARAMIS trial did not exclude patients with a history of seizures unlike the SPARTAN and PROSPER trials did. In the PROSPER, SPARTAN, and ARAMIS trials, the incidences of seizures in the placebo arms were 0%, 0%, and 0.2%, respectively, whereas those in the intervention (ENZA, APA, and DARO) arms were <1%, 0.2%, and 0.2%, respectively.

Looking at the results of the three studies, all three drugs showed better oncologic outcomes in nm CRPC than placebo did. However, there are no direct comparative studies between the three drugs. Therefore, it is not yet clear which drug is superior among them. Therefore, we tried to indirectly compare the results of the three drugs based on a published network meta-analysis of phase III RCT results [48,49,50,51,52].

### 4.1. MFS and PSA Progression-Free Survival

In the primary analyses, treatment with APA (HR, 0.28; 95% CI, 0.23–0.35; *p* < 0.001), ENZA (HR, 0.29; 95% CI, 0.24–0.35; *p* < 0.001), or DARO (HR, 0.41; 95% CI, 0.34–0.50; *p* < 0.001) than treatment with placebo as an adjunct to ADT was associated with an approximately two-year increase in median MFS. These results suggest the superiority in oncologic outcomes of the three drugs (APA, ENZA, and DARO) over placebo. According to a published network meta-analysis [49,50,51], the SPARTAN and PROSPER trials showed superiority over the ARAMIS trials in terms of MFS. However, there was no difference in the indirect comparison between APA and ENZA. Although PSA progression-free survival showed similar results, it is difficult to draw conclusions until a direct comparison is made. In a recently published study, DARO was compared with APA and ENZA by selection and reweighting to match the inclusion criteria and baseline characteristics of the patients; however, no statistically significant difference was seen in the MFS [53]. Therefore, a careful interpretation is required.

### 4.2. OS

In the final analyses, all RCTs showed an improvement in OS with the intervention compared with placebo; the SPARTAN, PROSPER, and ARAMIS trials reported HRs of 0.78 (95% CI; 0.64–0.96; *p* = 0.016), 0.73 (95% CI; 0.61–0.89; *p* = 0.001), and 0.69 (95% CI; 0.53–0.88; *p* = 0.003), respectively. A published network meta-analyses, analyses using the final data of each RCT, were performed by Roumiguie et al. [52]. However, current data are insufficient to assess the significant ranking of the three drugs over placebo in terms of OS. Unlike for MFS and PSA progression-free survival, all three studies showed similar results for OS. Although the ARAMIS trial has not yet reported the median OS, it showed similar OS results compared to a relatively low MFS. However, the OS may eventually differ when the drugs are administered after cancer progression. In the three studies, there were differences in subsequent therapy after nmCRPC progression. As for the choice of the drug used in the subsequent therapy, the SPARTAN trial used abiraterone more frequently than the other two did; the PROSPER trial used abiraterone and chemotherapy at a similar rate, whereas the ARAMIS trial prescribed chemotherapy the most often. The optimal subsequent therapy for CRPC remains controversial [54,55,56,57,58]. Currently, studies on biomarkers are underway to predict treatment response and prognosis. Among the biomarkers investigated, AR splice variant 7 (AR-V7) appears to be a promising biomarker for predicting the response to AR axis-targeted agents. AR-V7 is an important marker for resistance to AR axis-targeted agents [59]. There has been no prospective study of the AR-V7 mutation wherein secondary AR inhibitors were used. However, in this study, the mutation was slightly more when using ENZA or abiraterone. Although AR-V7 mutation is not an absolute criterion for drug use in patients with CRPC, it may be a factor in reducing the effectiveness of secondary ARi [60]. De Wit et al. reported that cabazitaxel showed better results than ENZA or abiraterone as subsequent therapy in mCRPC patients (HR, 0.64; 95% CI, 0.46–0.89; *p* = 0.008) [61]. Although chemotherapy cannot be considered in all patients [62,63], additional studies on the optimal subsequent therapy in nmCRPC patients are necessary.

### 4.3. Safety

The SPARTAN trial collected data on AEs at one-month intervals, whereas the PROSPER and ARAMIS trials collected AE-related data at four-month intervals. Compared to the placebo group, the APA group had higher rates of fatigue, HTN, rash, weight loss, arthralgia, and fracture, while the ENZA group had higher rates of fatigue, HTN, dizziness, falls, and fracture. The occurrence of fatigue was higher in the DARO group than in the placebo group.

There was an increase in HTN incidence in the SPARTAN and PROSPER intervention groups. It is a potent inducer of CYP3A4 due to the nature of the drug [64,65], which may be caused by interactions with antihypertensive drugs [66,67]. However, DARO has demonstrated a lower likelihood of drug–drug interactions than those of APA and ENZA because DARO is structurally distinct from the two drugs [46,68]. Cardiotoxicity is an important factor in the use of ENZA [69,70]. A previous meta-analysis showed a significant increase in the relative risk of all-grade and high-grade cardiac toxicity in patients receiving ENZA compared to that in patients receiving placebo [71]. In the PROSPER trial, cardiotoxicity increased by approximately 2% with ENZA than with placebo. APA had no cardiotoxic side effects apart from HTN, and DARO did not differ from placebo in terms of cardiotoxicity.

The SPARTAN, PROSPER, and ARAMIS trials reported the occurrence of grade 3–4 AEs among 55.9%, 48%, and 26.3% of the intervention (APA, ENZA, and DARO) group patients and 36.4%, 27%, and 26.3% of the placebo group patients, respectively.

In addition, as opposed to APA and ENZA, DARO did not have a higher rate of drug discontinuation due to AEs than placebo did. According to a network meta-analysis, DARO was found to be relatively superior to the two drugs in terms of safety.

As stated earlier, conclusions should not be drawn by comparing safety data from different trials because differences exist in clinical trial design and included populations. Ultimately, randomized head-to-head trials are required to compare safety profiles. Currently, the DaroAcT Trial (NCT04157088) is in progress and is expected to be completed in 2022 [72]; it directly compares DARO and ENZA to assess differences in physical and cognitive function.

### 4.4. Health-Related Quality of Life Outcomes

The ultimate treatment goal for nmCRPC is to maintain the patient’s QOL and delay time to metastasis. Therefore, each of the three clinical trials evaluated QOL using a verified questionnaire.

In the SPARTAN trial, health-related QOL (HR-QOL) was assessed using the Functional Assessment of Cancer Therapy–Prostate (FACT-P) and European Quality of Life (EQ) visual analog scale (VAS) [73,74]. After 29 months, for FACT-P, the APA and placebo groups reported mean scores of −0.99 ± 0.98 and −3.29 ± 1.97, respectively. Additionally, for EQ-VAS, the mean scores for the APA and placebo groups were 1.44 ± 0.87 and 0.26 ± 1.75, respectively. There was no statistical difference, but the APA group had slightly better QOL than did the placebo group. In the PROSPER trial, many comparisons were made between placebo and ENZA groups regarding QOL. The FACT-P total score for the ENZA group was significantly better than that for the placebo group (HR 0.83; 95% CI 0.69–0.99; *p* = 0.037). The mean score for the Brief Pain Inventory Short Form, which assesses pain severity, was reported to be better in the ENZA group than in the placebo group (HR 0.75; 95% CI 0.57–0.97; *p* = 0.028). In addition, patients showed better bowel symptoms, function, and urinary symptoms. The HR-QOL with DARO was reported based on preliminary data [35]. DARO significantly delayed pain progression (HR 0.65; 95% CI 0.53–0.79; *p* < 0.001) more than placebo did. Moreover, the delay in urinary symptoms was clinically significant with DARO (HR, 0.64; 95% CI, 0.54–0.76; *p* < 0.01) than with placebo. A recent study compared HR-QOL outcomes between APA and ENZA through matching-adjusted indirect comparisons. They reported that, based on FACT-P scores, APA showed better results than ENZA did [75]. However, since there is no direct comparison between the three drugs yet, it is difficult to evaluate which drug facilitates superior QOL. All three drugs may offer patients with nmCRPC a therapeutic option while maintaining QOL.

## 5. PSMA-PET Imaging in nmCRPC

Prostate-specific membrane antigen (PSMA) positron emission tomography (PET) imaging can detect metastatic disease in a significant proportion of nmCRPC patients, potentially leading to restaging. PSMA is overexpressed in prostate tumor cells. There are research results on restaging in nmCRPC patients using PSMA-PET [76,77,78,79]. Frendler et al. conducted a study using PSMA-PET by selecting patients similar to those included in the SPARTAN, PROSPER, and ARAMIS trials [80]. It was a multicenter, retrospective study of 200 patients with nmCRPC who had a PSA level >2 ng/mL, PSADT ≤10 months, and/or Gleason score ≥8. Conventional imaging showed absence of metastasis, but PSMA-PET showed positive findings in 98% of the patients. In 44% of the patients, the metastasis was confined to the pelvis with 24% of patients having metastasis in the prostate bed. Furthermore, metastasis was found in 55% of the patients who developed it in the extra pelvic nodes (39%), bone (24%), and visceral organs (6%). In another study, Fourquet et al. included a total of 30 patients from relatively low-risk groups [81]. Among them, PSMA-PET-positive findings were seen in 20 patients with PSA levels >2 ng/mL. In addition, 10 patients with PSA <2 ng/mL showed positive findings in 70%. PSMA-PET-positive lesions were confined to the prostate bed in 7% of patients. In addition, 20% of the patients had oligometastatic diseases with less than three lesions, and 63% of the patients had polymetastatic disease. The current definition of nmCRPC is based on conventional imaging modalities. However, given that PSMA-PET is a highly effective and universally available diagnostic tool, we believe that the definition of nmCRPC and treatment plan may change with the evolution of such diagnostic tools in the future. [82,83]

## 6. Conclusions

APA, ENZA, and DARO have excellent safety profiles for patients with nmCRPC. The HR-QOL was preserved and prostate cancer-related symptoms were significantly delayed in the intervention groups. Therefore, APA, ENZA, and DARO should be considered as novel standard therapies for nmCRPC. However, the effects of these three drugs should be compared through direct comparative studies in the future.

## Figures and Tables

**Table 1 biomedicines-09-00661-t001:** Comparison of basic characteristics for second-generation AR inhibitors.

	Apalutamide [18]	Enzalutamide [19]	Darolutamide [20]
Brand name	Erleada	Xtandi	Nubeqa
Dose/form	60 mg/tablet	40 mg/capsule	300 mg/film-coated tablet
Total daily dosage	240 mg (once per day)	160 mg (once per day)	600 mg (twice per day)
Route of administration	Oral administration	Oral administration	Oral administration
Approved indication	nmCRPC, mCSPC	nmCRPC, mCSPC, mCRPC	nmCRPC
Chemical structure	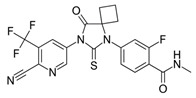	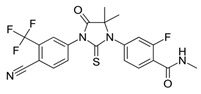	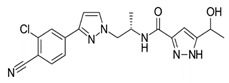
Bioavailability	100% [21]	Rats: 89.7% [22]; humans: unknown (at least 84.6%; based on recovery from excretion)	≤30%
Metabolites	N-Desmethylapalutamide	N-Desmethylenzalutamide (active) Carboxylic acid derivative metabolite (inactive)	Ketodarolutamide
Half-life	3–4 days (at steady state)	5.8 days (range 2.8–10.2 days)	16–20 h [23]
Excretion	Urine: 65% Feces: 24%	Urine: 71.0% Bile: 13.6% Feces: 0.39% [24]	Urine: 63.4% Feces: 32.4%
Blood–brain barrier penetration	Yes	Yes	Negligible
Mechanism of action for AR inhibition	It binds to the ligand-binding domain of the androgen receptor, blocks androgen-receptor nuclear translocation, inhibits DNA binding, and obstructs androgen receptor-mediated transcription.	It prevents the translocation of the AR from the cytoplasm to the nucleus. Within the nucleus, it inhibits AR binding to chromosomal DNA, which prevents further transcription of tumor genes.	It competitively inhibits androgen binding, AR nuclear translocation, and AR-mediated transcription. A major metabolite, keto-darolutamide, exhibited similar in vitro activity to darolutamide.

AR, androgen receptor; nmCRPC, non-metastatic castration-resistant prostate cancer; mCRPC, metastatic castration-resistant prostate cancer; mCSPC, metastatic castration-sensitive prostate cancer.

**Table 2 biomedicines-09-00661-t002:** Comparison of study populations and oncologic efficacy in nmCRPC treatment phase 3 clinical trials.

	Spartan [7,33] (NCT01946204)	Prosper [6,34] (NCT02003924)	Aramis [36] (NCT02200614)
Enroll patients	APA (*n* = 806) vs. PBO (*n* = 401)	ENZA (*n* = 933) vs. PBO (*n* = 468)	DARO (*n* = 955) vs. PBO (*n* = 554)
Inclusion criteria	M0N0-N1CRPC, PSADT < 10 months	M0N0CRPC, PSADT < 10 months, PSA >2 ng/mL	M0N0-N1CRPC, PSADT < 10 months, PSA >2 ng/mL
Median age (range)	APA 74 years (48−94) vs. PBO 74 years (52−97)	ENZA 74 years (50−95) vs. PBO 73 years (53−92)	DARO 74 years (48−95) vs. PBO 74 years (50−92)
Median follow-up (final)	52 months	48 months	29 months
Median PSA at baseline (ng/mL)	APA 7.78 vs. PBO 7.96	ENZA 11.1 vs. PBO 10.2	DARO 9.0 vs. PBO 9.7
Median PSADT (months)	APA 4.4 vs. PBO 4.5	ENZA 3.8 vs. PBO 3.6	DARO 4.4 vs. PBO 4.7
Median MFS (months)	APA 40.5 vs. PBO 16.2 24-month MFS benefit	ENZA 36.6 vs. PBO 14.7 22-month MFS benefit	DARO 40.4 vs. PBO 18.4 22-month MFS benefit
	72% reduction of distant progression or death; HR 0.28 (95% CI 0.23–0.35); *p* < 0.001	71% reduction of distant progression or death; HR 0.29 (95% CI 0.24–0.35); *p* < 0.001	59% reduction of distant progression or death; HR 0.41 (95% CI 0.34–0.50); *p* < 0.001
Final median OS (months)	APA 73.9 vs. PBO 59.9	ENZA 67.0 vs. PBO 56.3	Not reached
	22% reduction in risk of death HR 0.78 (95% CI 0.64–0.96); *p* = 0.016	27% reduction in risk of death HR 0.73 (95% CI 0.61–0.89); *p* = 0.001	31% reduction in risk of death HR 0.69 (95% CI 0.53–0.88); *p* = 0.003
Median time to PSA progression (months)	APA 40.5 vs. PBO 3.7	ENZA 40.5 vs. PBO 3.7	DARO 33.2 vs. PBO 7.3
	HR 0.07 (95% CI 0.06–0.09); *p* < 0.0001	HR 0.07 (95% CI 0.05–0.08); *p* < 0.001	HR 0.13 (95% CI 0.11–0.16); *p* < 0.001
Median PFS (months)	APA 40.5 vs. PBO 14.7	NR	DARO 36.8 vs. PBO 14.8
	HR 0.29 (95% CI 0.24–0.36); *p* < 0.0001		HR 0.38 (95% CI 0.32–0.45); *p* < 0.001
Time to initiation of cytotoxic chemotherapy	HR 0.63 (95% CI 0.49–0.81); *p* = 0.0002	NR	HR 0.58 (95% CI 0.44–0.76); *p* < 0.001
Time to initiation of suqsequent antineoplastic therapy	NR	HR 0.29 (95% CI 0.25–0.34); *p* < 0.001	HR 0.36 (95% CI 0.27–0.48); *p* < 0.001

APA, apalutamide; CI, confidence interval; DARO, darolutamide; ENZA, enzalutamide; HR, hazard ratio; MFS, metastasis-free survival, NR, not reported; OS, overall survival; PBO, placebo; PFS, progression-free survival; PSA, prostate-specific antigen; PSADT, PSA doubling time.

**Table 3 biomedicines-09-00661-t003:** Comparison of safety profile in nmCRPC treatment phase 3 clinical trials.

Safety	Spartan [7,33] (NCT01946204)		Prosper [6,34] (NCT02003924)		Aramis [13] (NCT02200614)	
	APA (*n* = 803)	PBO (*n* = 398)	ENZA (*n* = 930)	PBO (*n* = 465)	DARO (*n* = 954)	PBO (*n* = 554)
Any AE	781 (97)	373 (94)	876 (94)	380 (82)	818 (85.7)	439 (79.2)
Grade 3 or 4 AE	449 (56)	145 (36)	292 (31)	109 (23)	251 (26.3)	120 (21.7)
Any serious AE	290 (36)	99 (25)	372 (40)	100 (22)	249 (26.1)	121 (21.8)
AE leading to discontinuation	120 (15.0)	29 (7.3)	158 (17)	41 (9)	85 (8.9)	48 (8.7)
AE leading to death	24 (3.0)	2 (0.5)	51 (5.0)	3 (1.0)	38 (4.0)	19 (3.4)
Fatigue *	265 (33)	83 (21)	424 (46)	103 (22)	126 (13.2)	46 (8.3)
Hypertension *	225 (28)	83 (21)	161 (17)	27 (6)	74 (7.8)	36 (6.5)
Falls *	177 (22)	38 (9.5)	164 (18)	25 (5)	50 (5.2)	27 (4.9
Bone fracture *	145 (18)	30 (7.5)	168 (18)	29 (6)	52 (5.5)	20 (3.6)
Arthralgia *	160 (20)	33 (8.3)	119 (13)	36 (8)	86 (9.0)	52 (9.4)
Constipation *	not reported		121 (13)	39 (8)	66 (6.9)	36 (6.5)
Diarrhea *	184 (23)	60 (15)	112 (12)	47 (10)	71 (7.4)	31 (5.6)
Hot flush *	120 (15)	34 (8.5)	121 (13)	36 (8)	57 (6.0)	25 (4.5)
Mental impairment disorder *	41 (5.1)	12 (3)	73 (8)	10 (2)	19 (2.0)	10 (1.8)
Rash *	212 (26)	25 (6.3)	38 (4)	13 (3)	30 (3.1)	6 (1.1)
Seizure *	5 (0.6)	0	3 (<1)	0	2 (0.2)	1 (0.2)

AE, adverse events; APA, apalutamide; DARO, darolutamide; ENZA, enzalutamide; PBO, placebo; *, all grades of AE; data were presented as *n* (%).

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
