# Peer review of "Novel Treatment Strategy Using Second-Generation Androgen Receptor Inhibitors for Non-Metastatic Castration-Resistant Prostate Cancer"

_biomedicines, 2021, doi:10.3390/biomedicines9060661_

Round 1

Reviewer 1 Report

The work of Chung et coll. it is very well conducted according to international scientific criteria, on a topic of considerable clinical and oncological interest. The comparison between latest generation drugs is well represented. The tables could be graphically improved for a more immediate understanding. Furthermore, PSMA-PET is an established practice of nuclear medicine and not a new imaging modality; perhaps a PET image could have improved the quality of the work.

Reviewer 2 Report

Interesting paper that provides data of interest in relation to the efficacy of new treatments
